# Recent Advances in the Study of Trivalent Lanthanides and Actinides by Phosphinic and Thiophosphinic Ligands in Condensed Phases

**DOI:** 10.3390/molecules28176425

**Published:** 2023-09-04

**Authors:** Qin Wang, Ziyi Liu, Yu-Fei Song, Dongqi Wang

**Affiliations:** 1State Key Laboratory of Chemical Resource Engineering, Beijing Advanced Innovation Center for Soft Matter Science and Engineering, Beijing University of Chemical Technology, Beijing 100029, China; 2019400226@buct.edu.cn; 2State Key Laboratory of Fine Chemicals, Liaoning Key Laboratory for Catalytic Conversion of Carbon Resources, School of Chemical Engineering, Dalian University of Technology, Dalian 116024, China; wangdq@dlut.edu.cn; 3CAS-HKU Joint Laboratory of Metallomics on Health and Environment, Multidisciplinary Initiative Center, Institute of High Energy Physics, Chinese Academy of Sciences, Beijing 100049, China

**Keywords:** phosphinic ligands, lanthanides and actinides, liquid–liquid extraction, molecular dynamics

## Abstract

The separation of trivalent actinides and lanthanides is a key step in the sustainable development of nuclear energy, and it is currently mainly realized via liquid–liquid extraction techniques. The underlying mechanism is complicated and remains ambiguous, which hinders the further development of extraction. Herein, to better understand the mechanism of the extraction, the contributing factors for the extraction are discussed (specifically, the sulfur-donating ligand, Cyanex301) by combing molecular dynamics simulations and experiments. This work is expected to contribute to improve our systematic understanding on a molecular scale of the extraction of lanthanides and actinides, and to assist in the extensive studies on the design and optimization of novel ligands with improved performance.

## 1. Introduction

In recent years, there has been a transition of the energy strategy worldwide from non-renewable fossil fuels to renewable resources. This transition is also considered necessary in order to minimize the release of carbon dioxide to alleviate the greenhouse effect and environmental pollution. Nuclear energy is considered one of the most promising alternatives to fossil fuels. According to statistics, in 2022, the nuclear power plants in China generated 417.786 billion kW∙h, accounting for 4.98% of the total national power generation. This is equivalent to 118 million tons of standard coal, which is deemed to release substantial carbon dioxide, nitrogen oxides, and sulfur dioxide, demonstrating the advantage of nuclear energy as a highly efficient, clean, and sustainable energy solution.

With the growth of nuclear power generation, a large amount of nuclear waste needs to be disposed of [1,2]. The disposition of high-level waste in the reprocessing of spent fuel is key in the sustainable development of nuclear energy. The partitioning and transmutation strategy [3,4,5] offers a protocol to separate long-lived nuclides of minor actinides from spent fuel and to transform high-level waste to low-level waste for the safe geological disposal of nuclear waste [6]. These actinides have chemo- and radiotoxicity, and they can cause severe health consequences once taken up into the human body. However, lanthanide products co-exist in the spent fuel [7], and they are characterized by high neutron-absorption cross-sections. In order to guarantee the efficiency of transmutation, trivalent lanthanides should be removed from the spent fuel; however, it remains a challenge to separate lanthanides and minor actinides via established liquid–liquid extraction techniques owing to their similar ionic radii, as well as their physical and chemical properties in condensed phase. In the separation of trivalent lanthanides and actinides, the liquid–liquid extraction technique benefits from the more diffuse feature of the 5f orbitals of actinides than the 4f orbitals of lanthanides, which results in actinides being “softer” cations than lanthanide ions. According to the hard–soft acid–base (HSAB) principle [8,9], N- and S-donor ligands have higher affinity for An^3+^ than Ln^3+^; thus, they are considered to be potential ligands for the extraction and separation of An^3+^ from Ln^3+^. Many groups have improved the understanding of the solution dynamics of lanthanides, actinides, and their complexes with ligands toward the design and optimization of ligands for Ln/An separation via liquid–liquid extraction.

In this work, recent studies on the solution dynamics of phosphinic and thiophosphinic ligands and their complexes with lanthanide and actinide ions are reviewed to provide peer scientists with an understanding of the status of this issue.

## 2. Factors Influencing Dynamics of Ln^3+^ and An^3+^

The ligands used for Ln/An separation generally contain O, N, or S as ligating sites. The intrinsic conformational and electronic properties of the ligands determine their behavior in the extraction and exhibit different kinetics and selectivity. According to the hard–soft acid–base (HSAB) principle [8,9], the slightly softer An^3+^ has higher affinity for ligands containing N or S atoms than Ln^3+^; thus, these soft donor ligands exhibit selectivity for An^3+^ and can be used in the separation of An^3+^ from Ln^3+^. The ligands containing O atoms are commonly used for group extraction and chelation of lanthanides and actinides, but can hardly distinguish them.

The geometric structures of ligands govern their behavior in the condensed phase, including their coordination modes with metal ions and their migration in homogeneous and heterogeneous media. Many factors can influence the solution dynamics of lanthanide and actinide ions, such as the concentration of ligands, their protonation states (pH), and the type of solvent.

Owing to the difficulty in conducting experimental studies of actinides, molecular dynamics (MD) simulations have been employed to explore the extraction mechanism, including the key issues of, e.g., the coordination structure and the migration in the extraction. Below the recent advances in the understanding of the factors influencing the dynamics of actinides in the condensed phase are surveyed.

### 2.1. Concentration of Ligands

The concentration of ligands plays a potential role in the solution dynamics of ligands coordinating with lanthanides and actinides. Wipff and coworkers [10] demonstrated concentration-dependent distribution of tributyl phosphate (TBP, Figure 1) in the water/oil biphasic system using molecular dynamics simulations. TBP displayed a propensity to stay at the interface and, at low concentration, formed an unsaturated monolayer at the biphasic interface separating the aqueous phase and the oil phase, adopting an amphiphilic orientation. Increasing the concentration of TBP resulted in mixing of water and oil at the interface, which constituted a rough interface, and the orientations of TBP ligands at the interface were random with excess TBP molecules distributed in the organic phase. Bhattacharyya et al. [11] reported that Eu^3+^ and Cyanex301 (Figure 1) can form 1:2 complexes at a low extractant concentration (<0.3 M) and 1:3 complexes at a higher extractant concentration. This demonstrated the significant influence of ligand concentration on the coordination structure of the Eu^3+^–Cyanex301 complex.

### 2.2. Protonation State of Ligands (pH)

The acidity of the condensed phase (pH) governs the protonation state of ligands and the hydrolysis of metal ions, as well as affects their complexation mode and distributions and the complexes formed between them. Thus, the solution dynamics of lanthanides and actinides ions is strongly affected by the pH of solution. Wipff and coworkers [12] reported that neutral bistriazinylbipyridine (BTBP, as shown in as shown in Figure 1) was favorably distributed in the organic phase rather than at the interface, and protonated BTBPH^+^ ligands showed the propensity to assemble at the interface, making contact with water. According to the observed influence of protonation on the behavior of the ligand, the slow extraction kinetics observed experimentally was proposed to have two origins. Firstly, the neutral ligands with weak hydrophilicity are favorably dispersed in the organic phase and can hardly capture the metal ions at the interface or in the aqueous phase. Secondly, the protonated ligands may repel metal ions due to electrostatic repulsion, e.g., Eu^3+^, and prevent the complexion process in the absence of synergistic reagents. The neutral bistriazinylpyridine (BTP, as shown in Figure 1) is such a ligand with weak surface activity, and it can adopt multiple orientations at the interface according to molecular dynamics simulations [13], depending on time and their lateral and para substituents. Its protonated forms show, in contrast, a strong propensity to distribute at the interface to interact with water.

Wipff et al. [14] also explored the solution dynamics of Cyanex301-like ligands (Figure 1) in extracting Eu^3+^ using molecular dynamics simulations, and both their intact and deprotonated forms were surface-active and populated at the interface, with the latter being more surface-active. Dwadasi and coworkers [15] reported that the protonation of phosphoric ligands played a pivotal role in the adsorption of rare earth elements ions (Nd^3+^ and Dy^3+^) at the interface, forming a number of different complexes at the interface involving one to three extractant molecules and four to eight aqua ligands bound.

In summary, the protonated states of ligands have an important effect on the distribution and behavior in the water/oil biphasic systems.

### 2.3. Types of Co-Existing Ions

Ion species co-present in the condensed phase can bring a pronounced effect on the ability of ligands to capture lanthanide and actinide ions. This is a consequence of the intrinsic properties of the ion species.

Firstly, the ions have different interfacial propensity. In an extraction study [16] of Lu^3+^ by dihexadecyl phosphate (DHDP, Figure 2) using surface-specific X-ray reflectivity (XR), X-ray fluorescence near-total reflection (XFNTR), and vibrational sum frequency generation (VSFG) spectroscopy techniques, Uysal et al. reported that the interface was occupied by negatively charged ligands, and SCN^−^ anions were also distributed at the interface to enhance the nucleophilicity of the interface, which was beneficial for capturing Lu^3+^. In the same year, Uysal and coworkers [17] extracted heavy lanthanides with different types of salts. SCN^−^ ions preferred to distribute in the interface than NO_3_^−^ and mess up the interface according to XR, grazing incidence X-ray diffraction (GID), and VSFG spectroscopy techniques, thus possibly promoting the extraction of heavy lanthanide ions through electrostatic interactions.

Secondly, the ions display different salting-out properties. The salting-out effect generally appears when adding inorganic salts to the aqueous medium, which reduces the solubility of a substance and results in its precipitation; this is suggested to be one of the important factors influencing metal extraction [18,19]. In a combined experimental and MD study of extraction of Pr^3+^ by trioctylphosphine oxide (TOPO, Figure 2) [20], Sun et al. reported that, at low enough concentrations of coexisting salts in aqueous solutions, the extraction of Pr^3+^ was mainly dominated by the interfacial propensity of those anions but not their salting-out ability, and SCN^−^ had the strongest ability to attract Pr^3+^ to the interface. With the increase in the concentration of salts, the salting-out effect gradually became significant and, therefore, co-dominated the transport of Pr^3+^ ions across the liquid/liquid interface. Among the anions, NO_3_^−^ ions showed higher salting-out ability than ClO_4_^−^ and SCN^−^ ions, and, when their concentrations went up to a certain value, the extraction of Pr^3+^ ions induced by NO_3_^−^ ions was further enhanced compared to that induced by ClO_4_^−^ and SCN^−^ ions.

Thirdly, the hydration of anions has an important effect on the extraction. In an experimental study [21], the system with NO_3_^−^ showed a propensity to extract light lanthanides ions, while the system with SCN^−^ preferred to extract heavy lanthanides ions, benefitting from its rich distribution at the interface and greater accessibility to the positive monolayer constituted by metal cations. The electrostatic interactions dominated in the extraction, and heavy lanthanides with higher electron density were preferably extracted.

Lastly, the ability of ligands to coordinate with metal ions influences the extraction. In 2020, Xiao et al. [22] studied the extraction of lanthanides using the tetradentate phenanthroline-derived phosphonate (POPhen) ligand, C4-POPhen (Figure 3), by solvent extraction, NMR titration, UV/Vis titration, and single-crystal X-ray diffraction techniques, and they explored the effects of Cl^−^, NO_3_^−^, and ClO_4_^−^ anions. The results showed that, in the presence of ClO_4_^−^ anion, C4-POPhen ligands displayed excellent extraction capacity and selectivity for the heavy lanthanides, Lu^3+^, benefitting from the weaker coordinated ability of the ClO_4_^−^ anion than the other two counterions. Both 1:1 and 1:2 Lu(III)/C4-POPhen complexes were formed in all three systems, and the system with ClO_4_^−^ counterions preferred to form 1:2 Lu(III)/C4-POPhen species. In a combined experiment and computational study of the extraction of La^3+^ and Y^3+^ by D2EHPA (Figure 3) [23], the molecular dynamics simulations revealed that NO_3_^−^ formed a 1:1 complex with La^3+^ ([LaNO_3_(H_2_O)_7_]^2+^) and 1:2 complex with Y^3+^ ([Y(NO_3_)_2_(H_2_O)_7_]^+^). The slope analysis studies in the laboratory, combined with the molecular dynamics simulations, revealed that two and one ligands from the D2EHPA phosphoric acid extractant were required for the complete extraction of lanthanum and yttrium ionic complexes, respectively.

### 2.4. Type of Dilute Phase (Organic Phase)

The dilute phase used in the liquid–liquid extraction provides a medium to accommodate the ligand-bound metal ion species. Wipff et al. [10] investigated chloroform and supercritical CO_2_ (SC-CO_2_) in the extraction of uranyl ion using molecular dynamics simulations. The most significant difference between the two types of media was the lower solubility of water in chloroform and the higher mobility of the molecules in the SC-CO_2_.

Wipff and co-workers [13] explored the effect of the dilute phase on the extraction of Eu^3+^ by BTP solvated in hexane + octanol, hexane, nitrobenzene, or chloroform using molecular dynamics simulations. BTP dispersed well in chloroform and nitrobenzene, but aggregated in hexane and hexane + octanol. In the biphasic systems of water with the former two solvents, BTP was distributed at the interface with its molecular plane parallel with the interface; in the biphasic system of water with the latter, at the interface, BTP could orientate itself with its molecular plane parallel to the interface or with its nitrogen atoms pointing toward the dilute phase. Liu et al. [24] added octanol molecules into dodecane, resulting in the enhancement of the interfacial activity of ligands and benefitting the migration of ligands to the organic phase, as revealed by molecular dynamics simulations. The findings indicated that the polarity of the oil phase could contribute to the distribution of ligands in biphasic systems.

Ionic liquids, as “green” solvents with low volatility, high thermal stability, and wide liquid temperature range, can be used in the liquid–liquid extraction of lanthanide and actinide ions, with potential application prospects [25,26,27,28,29,30,31,32,33]. Wipff et al. [34] combined experiments and molecular dynamics simulations to explore the complexes of uranyl in ionic liquids, and found that nitrate ions and chloride ions could compete to coordinate with uranyl ions. Jiang and co-workers [35] simulated the dynamics of the preorganized 1,10-phenanthroline-2,9-dicarboxamide ligand (L) bound with Am^3+^ cation in the butylmethylimidazolium bistriflimide ([BMIM][NTf_2_]) ionic liquid. For both Am:L (1:1) and Am:L_2_ (1:2) complexes, it was found that the secondary solvation environment was influenced by the imidazolium arms of the ligands attracting NTf_2_ anions near the metal ion. As a result, the binding free energy for the second ligand was twice that for the first ligand, which resulted in the Am:L_2_ complex being more stable than the Am:L complex. The preorganized ligands with charged functional groups could be tuned to enhance the selectivity in ion extraction efficiency.

## 3. Phosphinic Ligands Bound Lanthanides and Actinides

Efficient separation of trivalent lanthanides and actinides is the key to establishing an advanced nuclear fuel cycle, which remains a challenge owing to their similar chemical and physical properties in the condensed phase [36]. According to the hard–soft acid–base (HSAB) principle, both trivalent lanthanides and actinides can be looked as “hard” Lewis acids with the latter being a little “softer” than the former due to the more diffuse 5f orbital of actinides. This constitutes the foundation of established liquid–liquid extraction protocols to separate trivalent lanthanides and actinides using soft donor (sulfur and nitrogen atoms) ligands [37,38,39,40,41,42,43,44,45].

As potential candidates to separate actinides and lanthanides, sulfur donor ligands display high selectivity for actinides over lanthanides, and the separation factor (SF) of An/Ln in the liquid–liquid extraction can be up to, depending on the dilute phase and the pH of the aqueous phase, SF = 5900 (kerosene, pH 2.8~3.4) [42], 6000 (dodecane, pH 3.4) [46], and 9800 (toluene) [47] for Cyanex301, and SF = 100,000 [48] for bis(o-trifluoromethylphenyl)dithiophosphinic acid (Figure 4). These values are much higher than those of nitrogen donor ligands (SF commonly around 200 [49]; maximal value of 1620 reported for CyMe4BTP [50]). Compared to N-donor ligands, softer sulfur ligands are less studied due to their deficiency, e.g., the sulfur pollution of high-level waste [51], their low radio-resistance [52,53], and the difficulty in their synthesis and purification [47,48,54].

Bis(2,4,4-trimethylpentyl) dithiophosphinic acid (Cyanex301) is a representative sulfur donor ligand to selectively separate lanthanides and actinides. In 1995, Zhu et al. [55] selectively extracted Am^3+^ from trivalent lanthanides by Cyanex301; in the next year, they reported that the SF value of Am^3+^ and Eu^3+^ could be up to 5900 [42]. In 1998, Modolo and Odoj [53] reported that very high An(III)/Ln(III) separation factors (higher than 10^3^) were obtained for micro and macro concentrations of Ln(III) at pH 3–4. In 2002, Chen et al. [56] used purified Cyanex301 to separate 99.93% Am^3+^ from lanthanides.

The growing attention toward the potential of Cyanex301 to be used in the selective separation of trivalent lanthanides and actinides calls for an extensive understanding of the origin of its selectivity, which remains controversial. There are three main opinions, emphasizing the coordinating structures of complexes, the dehydration of lanthanides and actinides, or the covalency of the coordinated bonds. Below, we briefly survey the studies related to these three opinions.

### 3.1. The Coordinated Structures of the Complexes

In 2002, Jensen et al. [57] found that Cyanex301 ligands could form neutral bidentate ML_3_ complexes with Nd, Sm, and Cm using visible absorption spectroscopy and X-ray absorption fine structure (XAFS) measurements, as shown in Figure 5. In 2003, Tian et al. [58] reported different structures of complexes of Cyanex301 with lanthanides (La^3+^, Nd^3+^, and Eu^3+^) and actinides (Am^3+^) (Figure 5), whereby Ln^3+^ was coordinated by seven sulfur atoms from ligands and one oxygen atom from water, HML_4_∙H_2_O (L = C301), while eight sulfur atoms from four Cyanex301 ligands were coordinated with Am with no aqua ligand, HML_4_ (L = C301). In 2007, Bhattacharyya et al. [11] reported the influence of the concentration of Cyanex301 on the structures of the Am^3+^ and Eu^3+^ complexes with Cyanex301 in nitric acid. At a low concentration of Cyanex301 (less than 0.3 mol/L), the coordinated complexes took the form of AmL_3_ and EuL_2_NO_3_ (L = C301), respectively. In the concentrated Cyanex301 systems, the complex of Eu^3+^ changed into Eu(C301)_3_.

These results show that the coordinated structures of lanthanides and actinides extracted by Cyanex301 are different and influenced by the concentration of Cyanex301 and the property of the solvent, resulting in a complicated mechanism underlying the distinctly different selectivity of Cyanex301 for lanthanides and actinides.

### 3.2. The Dehydration of Lanthanides and Actinides

When using Cyanex301 to extract lanthanides and actinides, depending on the properties of the metal ions, both inner-sphere complexes (Figure 6a) and outer-sphere complexes or microemulsions (Figure 6c) can form. In the outer-sphere complexes, the first coordinated shell of metal ions is occupied by water ligands, and the Cyanex301 ligands are distributed in the outer layer. In an EXAFS study, Tian et al. [59] reported that light lanthanides were mainly coordinated by sulfur atoms of Cyanex301, middle lanthanides were coordinated by sulfur atoms of Cyanex301 and oxygen atoms of aqua ligands, and heavy lanthanides were fully coordinated by aqua ligands in the first coordination shell and deprotonated Cyanex301 anions located in the outer shell to neutralize the positive charge.

In 2016, Chen et al. [60] investigated the influence of the neutralization of the system by NaOH on the structures of the complexes in the extraction of Nd^3+^ by Cyanex301, and they found that, when 10% of HC301 was neutralized, the inner coordination shell of Nd^3+^ in the organic phase was occupied by sulfur atoms from Cyanex301 and there was no formation of a W/O microemulsion. As the system was further neutralized, more HC301 was deprotonated, and a W/O microemulsion could be observed in the system. In a follow-up study, they [61] found that all lanthanides could form microemulsions in the extraction by Cyanex301. At a low deprotonation of Cyanex301, the heavy lanthanides could form microemulsions, and the light lanthanides were coordinated directly by Cyanex301 ligands. They [62] also reported that heavy lanthanides (Gd–Lu) preferred to form outer-sphere complexes with Cyanex301, whereas light lanthanides (La–Eu) displayed a propensity to form inner-sphere complexes.

Cao and coworkers [63] investigated the neutral complexes ML_3_ (M = Eu, Am, Cm; L = C301) by density functional theory, and reported that the complex with C301 as a bidentate ligand and the metal cation coordinated by six sulfur atoms were likely the most stable extraction complexes. They proposed that the higher selectivity of Cyanex301 for actinides may originate from the different hydration Gibbs free energies with the hydration Gibbs free energies of Eu^3+^ being higher than those of Am^3+^ and Cm^3+^, resulting in the complex of Eu^3+^ being less stable in the aqueous solution. Similar results were reported in a recent density functional theory study of the coordination of Eu(III) and An(III) with N-donor ligands, and their distinct ability to be dehydrated was proposed to govern their selective separation in liquid–liquid extraction [64].

### 3.3. The Covalency of Coordinated Bonds

First-principles studies have attributed the selectivity of Cyanex301 for actinides to the covalent feature of the interaction between them. In 2011, Bhattacharyya et al. [65] found that the higher covalency of the Am–S bond resulted in the selectivity of Cyanex301 for actinides according to density functional theory. Moreover, they reported that the solvent can influence the selectivity of Cyanex301. In 2014, Xu and coworkers [66] proposed that, in addition to the extent of covalency, the degree of desolvation and the coordination modes also contributed to differentiating the selective binding of Cyanex301 to Cm^3+^ over Nd^3+^. In a later density functional theory study, Cyanex301 was identified to combine with Am^3+^/Cm^3+^ to form ML_3_ complexes (L = C301), and the binding strength was stronger than with Eu^3+^ ions [67]. The extent of covalency was also considered as the origin of the different coordination modes of S- and O-containing ligands with lanthanide series [68], which became larger as the atomic number of lanthanides increased according to theoretical calculations, leading to the formation of both inner- and outer-sphere complexes for light lanthanides and only outer-sphere complexes for heavy lanthanides.

In addition to the controversy on the origin of the selectivity of Cyanex301, the mechanism of extraction of lanthanides and actinides by Cyanex301 is under debate. At present, the widely accepted extraction mechanism is the cation-exchange mechanism [30], i.e., where the metal ions exchange with protons.

As discussed above, liquid–liquid extraction is a complicated process in the condensed phase, which can be influenced by multiple factors, e.g., the aggregate of ligands, the ratio of metal ions and ligands in the complexes, the saponification of ligands, and the dilute phase. In practice, two methods are commonly used to improve the extraction of lanthanides and actinides by phosphinic ligands, i.e., the modification of the substituent and the addition of synergistic reagent to the dilute phase.

The substituents can affect the hydrophilicity and hydrodynamics of the ligands, which impose an influence on the extraction ability of the ligands. In 2013, Chen et al. [69] reported that Cyanex301 with shorter-branched alkyl chains showed improved extraction and separation ability for Am^3+^ and Eu^3+^ compared to native Cyanex301. In a later study [70], the –CF_3_ substituent was introduced into dithiophosphinic acids (DPAH, Figure 4), and the extraction ability of the ligands was found to be enhanced. A density functional theory study indicated that the higher selectivity for Am^3+^ than Eu^3+^ could be attributed to the stronger covalency of Am–S bonds. In another combined experimental and computational study by the same group [71], the ligands with *o*-CF_3_ were found to have a stronger interaction with Nd^3+^, and the improved ability to extract Nd^3+^ ions was proposed to be due to steric hindrance of the *o*-CF_3_ substituent, which benefited the dehydration of metal ions. Xu et al. [72,73] modified phosphinic acid by replacing its alkyl chains with unbranched alkyl chains, branched alkyl chains, and aromatic groups to explore the effects of substituents on the extraction of Am^3+^ and Eu^3+^; they found that the branched alkyl chain substituent and aromatic substituent exhibited better selectivity in the separation of Am^3+^ and Eu^3+^, which was attributed to the enhancement of the acidity of the ligands [47,72,74].

In liquid–liquid extraction, it is a common operation to add synergistic reagents to improve the extraction. Hill et al. [46] reported that the copresence of Cyanex301 and TBP could separate Am^3+^ and Eu^3+^ more efficiently, with the SF being up to 6000 higher than the value when using only Cyenax301. The difference may mainly originate from the different coordinated structures of the extracted complexes, as shown in Figure 7. Such enhancement was also reported by Torkaman et al. [75], who reported that the synergistic extraction of Sm^3+^ by Cyanex301 and D2EHPA performed better than when only using Cyanex301, as well as by Ionova et al. [76], who evaluated the synergistic extraction by Cyanex301 and some neutral oxygen-containing ligands, and reported a liner correlation of the D value of Am^3+^ and Eu^3+^ with the effective charge of oxygen atoms in the neutral oxygen-containing ligands.

At present, synergistic extraction is a potential way to improve the separation of lanthanides and actinides. However, the underlying mechanism is complicated and remains to be clarified to assist in the optimization of synergistic extraction in a specific extraction system.

## 4. Conclusions

To achieve the sustainable development of nuclear energy and eliminate the concerns regarding its threat to human beings, it is of significance to understand the extraction of ligands and ligands coordinating with lanthanides and actinides. Soft sulfur-donating ligands display significant selectivity for the trivalent actinides compared to lanthanides, and they are considered among the potential extractants in Ln/An separation. In order to provide a systematic view of the state of the art to promote extensive studies, we summarized herein the contributing factors that influence the solution dynamics and the extraction of trivalent lanthanides and actinides, i.e., the concentration of ligands, the pH, the type of salt ion, the type of dilute phase, and the specifics revealed by studies of sulfur-containing ligands.

Among these sulfur-donating ligands, Cyanex301 is a successful sulfur-based extractant and shows good performance in the separation of trivalent actinides and lanthanides. This review discussed the possible origins of selectivity of Cyanex301 for the actinides to contribute to an understanding of the mechanism underlying the selectivity of Cyanex301 and to assist in the design of Cyanex301 analogues with optimal performance for the extraction of actinides.

This work is expected to enrich our understanding of the extraction and the dynamics of ligands combined with actinides and lanthanides, and to optimize the extraction of lanthanides and actinides.

## Figures and Tables

**Figure 1 molecules-28-06425-f001:**
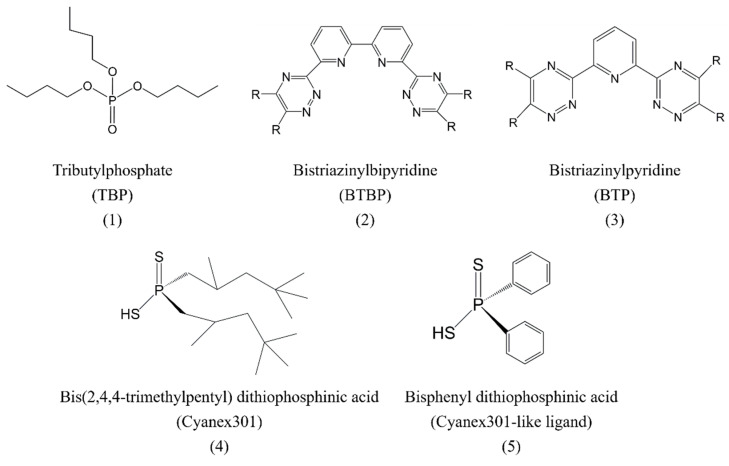
Schematic elucidation of the structures of TBP (**1**), BTBP (**2**), BTP (**3**), Cyanex301 (**4**), and Cyanex301-like ligand (**5**).

**Figure 2 molecules-28-06425-f002:**
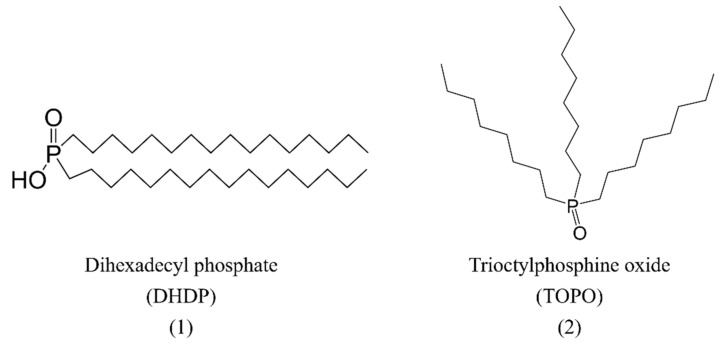
Structures of DHDP (**1**) and TOPO (**2**).

**Figure 3 molecules-28-06425-f003:**
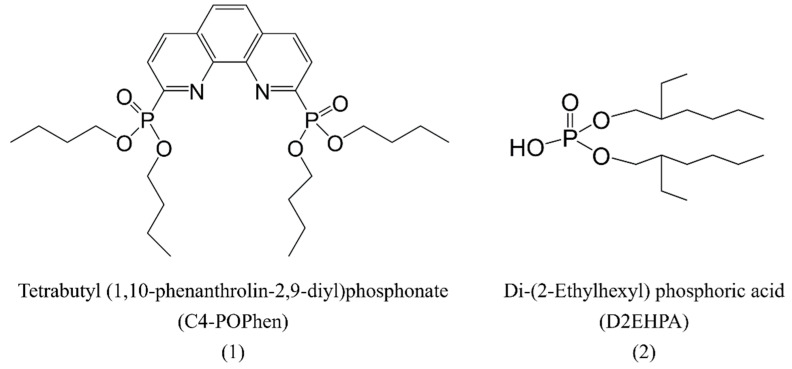
Structures of C4-POPhen (**1**) and D2EHPA (**2**).

**Figure 4 molecules-28-06425-f004:**
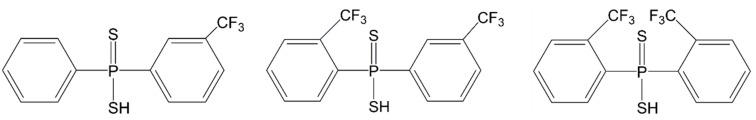
The three types of structures of trifluoromethyl substituted aryl dithiophosphinate.

**Figure 5 molecules-28-06425-f005:**
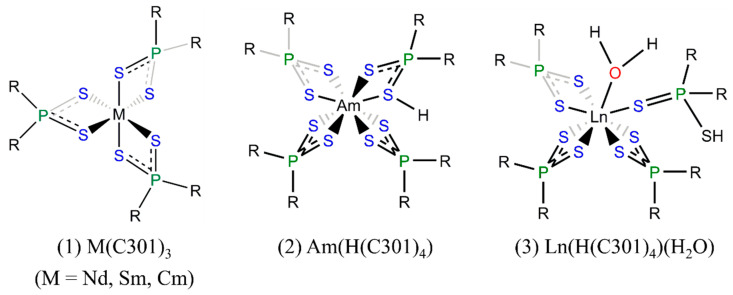
Structures of complexes in extracting lanthanides and actinides by Cyanex301 [57,58].

**Figure 6 molecules-28-06425-f006:**
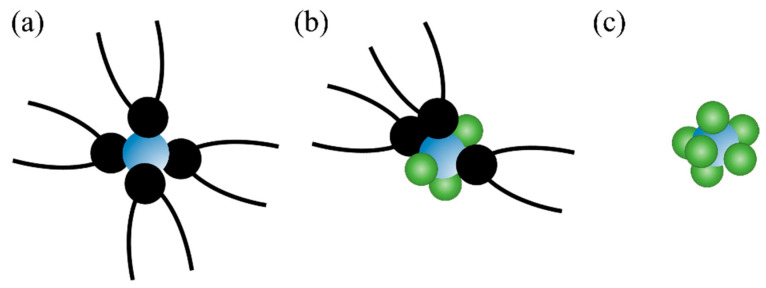
The different structures in the first coordinated shell of metal ions: (**a**) coordinated by organic ligands, (**b**) coordinated by organic ligands and water molecules, (**c**) coordinated by water molecules. Color scheme: green for water molecules (monodentate), black for coordinated ligands (bidentate), and blue for metal ions.

**Figure 7 molecules-28-06425-f007:**
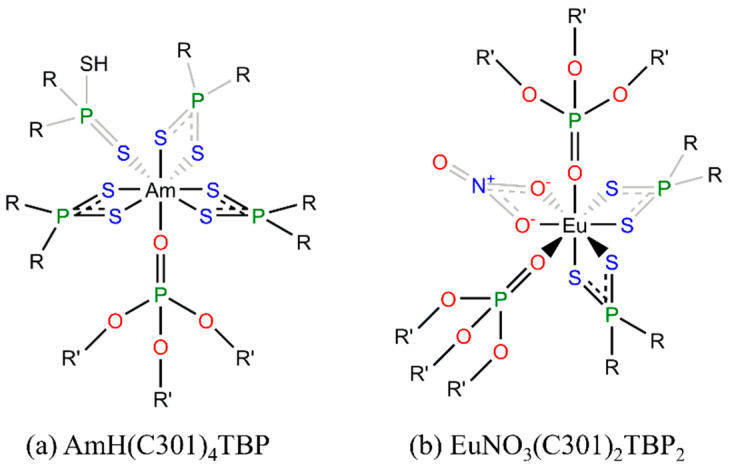
Structures of complexes in extracting (**a**) Am^3+^ and (**b**) Eu^3+^ by Cyanex301 and TBP (R = 2,4,4-trimethylpentyl, R’ = n-butyl).

## Data Availability

Not applicable.

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
