# Peer review of "Recent Advances in the Study of Trivalent Lanthanides and Actinides by Phosphinic and Thiophosphinic Ligands in Condensed Phases"

_molecules, 2023, doi:10.3390/molecules28176425_

Round 1
Reviewer 1 Report
Radioactive nuclear waste treatment is getting more attention as Japan plans to dump wastewater from Fukushima nuclear meltdown into the Pacific Ocean. Wang and coauthors reviewed the recent studies on the solution dynamics of phosphinic ligands and their complexes with Lanthanides and Actinides ions. In my opinion this work is very interesting especial they proposed that the Cyanex301 is a successful sulfur based extractant and has a good performance in the separation of trivalent actinides and lanthanides. Therefore it can be published in molecules after a minor revision based on the following comments:
Special comments:
1. Please indicate the line number of the submitted version.
2. In the Introduction section, some recent/relevant references are recommended for your consideration, such as: Separation and Purification Technology (2023), 325, 124662; J. Mol. Liq. (2021), 344, 117687; which might be helpful for readers to better understand the background and advances of the related area in this study.
3. There are some mistakes in the references. For example, the journal abbreviation in reference 18 and the random code in reference 36 and 37. Please check and amend it.
4. Please carefully check the citation format of the references. If references are inserted in the text, they should be placed before the punctuation.
Author Response
Radioactive nuclear waste treatment is getting more attention as Japan plans to dump wastewater from Fukushima nuclear meltdown into the Pacific Ocean. Wang and coauthors reviewed the recent studies on the solution dynamics of phosphinic ligands and their complexes with Lanthanides and Actinides ions. In my opinion this work is very interesting especial they proposed that the Cyanex301 is a successful sulfur based extractant and has a good performance in the separation of trivalent actinides and lanthanides. Therefore it can be published in molecules after a minor revision based on the following comments:
Special comments:
- Please indicate the line number of the submitted version.
Response1: We acknowledge the suggestion of the reviewer. The line number of the submitted version has been indicated.
- In the Introduction section, some recent/relevant references are recommended for your consideration, such as: Separation and Purification Technology (2023), 325, 124662; J. Mol. Liq. (2021), 344, 117687; which might be helpful for readers to better understand the background and advances of the related area in this study.
Response2: We acknowledge the reviewer for bringing the two papers to our attention. The two references have been cited as Ref. 1 and Ref. 2 in the manuscript.
- There are some mistakes in the references. For example, the journal abbreviation in reference 18 and the random code in reference 36 and 37. Please check and amend it.
Response3: We acknowledge the careful reading of the reviewer. The mistakes in the references have been checked and amended.
- Please carefully check the citation format of the references. If references are inserted in the text, they should be placed before the punctuation.
Response4: We acknowledge the suggestion of the reviewer. The citation format of the references has been checked and modified.

Reviewer 2 Report
This review paper deals with very important issues, such as the processes of separation of lanthanides and actinides, which are strategic elements of the 21st century.
As this is a review article, its substantive content does not raise my objections. I have a remark about the introduction part, which is too laconic. The authors should give a more general description of these two groups of elements and indicate their main directions of application. Moreover, the work should include a brief explanation of the extraction process of these elements, because not all readers of the Journal are well versed in this methodology.
Do only nuclear power plants generate waste containing these separated elements or are there other sources as well.? If so, please include this information in your work.
Author Response
This review paper deals with very important issues, such as the processes of separation of lanthanides and actinides, which are strategic elements of the 21st century.
As this is a review article, its substantive content does not raise my objections. I have a remark about the introduction part, which is too laconic. The authors should give a more general description of these two groups of elements and indicate their main directions of application. Moreover, the work should include a brief explanation of the extraction process of these elements, because not all readers of the Journal are well versed in this methodology.
Do only nuclear power plants generate waste containing these separated elements or are there other sources as well.? If so, please include this information in your work.
Response: We acknowledge the suggestion of the reviewer. A few sentences about the two groups of elements and the extraction process were added to the “Introduction” section:
These actinides have chemo and radiotoxicity, and can cause severe health consequences once uptaken into human body. However, there are lanthanide products co-existed in the spent fuel[7], which have high neutron-absorption cross sections. In order to guarantee the efficiency of transmutation, the trivalent lanthanides should be removed from the spent fuel, which remains a challenge to separate lanthanides and minor actinides via established liquid-liquid extraction techniques owing to their similar ionic radii, physical and chemical properties in condensed phase. In the separation of trivalent lanthanides and actinides, the liquid-liquid extraction technique benefits from the more diffuse feature of the 5f orbitals of actinides than the 4f orbitals of lanthanides, which results in actinides being “softer” cations than lanthanide ions. According to the hard-soft acid-base (HSAB) principle [8, 9], the N- and S-donor ligands have higher affinity for An3+ than Ln3+, and have been considered to be potential ligands to extract and separate An3+ from Ln3+.
This is currently the problem in the civil application of nuclear energy and a key component in the development of advance spent fuel reprocessing for sustainable use of nuclear energy.

Reviewer 3 Report
I believe the topic of this paper is interesting to general readers and it also fits well to the scope of the journal. The manuscript could be further improved if the following points is taken into account.
1. The title specifically refers to phosphinic ligands. The paper, on the other hand, puts heavy focus on Cyanex301 which is a sulfur donor. Also the conclusions says “… soft sulfur donating ligands display significant selectivity for the trivalent actinides comparing to lanthanides and have been considered among the choices of extractants in the Ln/An separation.” Whereas the title is about the phosphorous-containing ligand, conclusion is exclusively about coordination by sulfur. This gap needs to be resolved.
2. The subsection 2.1 exclusively focuses on uranyl system, which is not a trivalent ion. So this example does not fit to the topic of the paper. The authors could introduce more relevant example(s) involving Ln3+ and/or An3+.
3. I am not convinced about the structures depicted in Fig.5, especially the one for Am(III). It is highly unlikely that same four ligands are bound to the metal and only one of them has protonated sulfur. References 50 and 51 are not appropriate source to cite these hypothetical structures (I did not find these structures in the paper).
I am not in a position to comment on this.
Author Response
I believe the topic of this paper is interesting to general readers and it also fits well to the scope of the journal. The manuscript could be further improved if the following points is taken into account.
- The title specifically refers to phosphinic ligands. The paper, on the other hand, puts heavy focus on Cyanex301 which is a sulfur donor. Also the conclusions says “… soft sulfur donating ligands display significant selectivity for the trivalent actinides comparing to lanthanides and have been considered among the choices of extractants in the Ln/An separation.” Whereas the title is about the phosphorous-containing ligand, conclusion is exclusively about coordination by sulfur. This gap needs to be resolved.
Response1: We acknowledge the suggestion of the reviewer. The title was modified to mention the thiophosphinic ligands as below:
Recent Advances in the Study of Trivalent Lanthanides and Actinides by Phosphinic and Thiophosphinic Ligands in Con-densed Phases
- The subsection 2.1 exclusively focuses on uranyl system, which is not a trivalent ion. So this example does not fit to the topic of the paper. The authors could introduce more relevant example(s) involving Ln3+ and/or An3+.
Response2: We acknowledge the suggestion of the reviewer. The relevant content about uranyl system has been removed.
- I am not convinced about the structures depicted in Fig.5, especially the one for Am(III). It is highly unlikely that same four ligands are bound to the metal and only one of them has protonated sulfur. References 50 and 51 are not appropriate source to cite these hypothetical structures (I did not find these structures in the paper).
Response3: We acknowledge the suggestion of the reviewer. The mentioned References 50 and 51 in Fig.5 were mis-cited, and have been corrected as Ref57 (J. Am. Chem. Soc. 2002, 124, 9870-9877) and Ref58 (Inorg. Chem. 2003, 42, 735-741). In Ref58, Am3+ was reported to coordinated to four ligands, as shown in below excerpted sentences and the table.
